# Application of Ultrasound Scores (Subjective Assessment, Simple Rules Risk Assessment, ADNEX Model, O-RADS) to Adnexal Masses of Difficult Classification

**DOI:** 10.3390/diagnostics13172785

**Published:** 2023-08-28

**Authors:** Mar Pelayo, Javier Sancho-Sauco, Javier Sánchez-Zurdo, Belén Perez-Mies, Leopoldo Abarca-Martínez, Mª Jesús Cancelo-Hidalgo, Jose Antonio Sainz-Bueno, Juan Luis Alcázar, Irene Pelayo-Delgado

**Affiliations:** 1Universitary Hospital HM Puerta del Sur, HM Rivas, 3428521 Madrid, Spain; mar_pelayo@yahoo.es; 2Department of Obstetrics and Gynecology, Universitary Hospital Ramón y Cajal, Alcalá de Henares University, 3428034 Madrid, Spain; jsanchosauco@gmail.com (J.S.-S.); leohrcfetal@gmail.com (L.A.-M.); 3Héroux-Devtek, 3428906 Madrid, Spain; javier.sanchez@ciorank.com; 4Department of Pathology, Universitary Hospital Ramón y Cajal, Alcalá de Henares University, 3428034 Madrid, Spain; bperezmies@gmail.com; 5Department of Obstetrics and Gynecology, Universitary Hospital of Guadalajara, Alcalá de Henares University, 3428034 Madrid, Spain; mjesus.cancelo@uah.es; 6Department of Obstetrics and Gynecology, Valme Universitary Hospital, 3441014 Seville, Spain; jsainz@us.es; 7Department of Obstetrics and Gynecology, Clínica Universidad de Navarra, 3431008 Pamplona, Spain; jlalcazar@unav.es

**Keywords:** transvaginal ultrasound, adnexal masses, IOTA Simple Rules Risk Assessment, O-RADS, ADNEX model, CA125, subjective assessment, mature cystic teratoma, serous cystadenoma, mucinous cystadenoma, Brenner tumour, cystadenofibroma, fibroma, borderline carcinoma, serous ovarian carcinoma, clear cell carcinoma, endometriod carcinoma

## Abstract

**Featured Application:**

**Ultrasound scores should consider that some frequent masses such as fibromas, cystoadenofibromas, some mucinous cystadenomas and Brenner tumors may present some characteristics that induce confusion with malignant lesions. Some malignant lesions are not always identified as malignant.**

**Abstract:**

Background: Ultrasound features help to differentiate benign from malignant masses, and some of them are included in the ultrasound (US) scores. The main aim of this work is to describe the ultrasound features of certain adnexal masses of difficult classification and to analyse them according to the most frequently used US scores. Methods: Retrospective studies of adnexal lesions are difficult to classify by US scores in women undergoing surgery. Ultrasound characteristics were analysed, and masses were classified according to the Subjective Assessment of the ultrasonographer (SA) and other US scores (IOTA Simple Rules Risk Assessment-SRRA, ADNEX model with and without CA125 and O-RADS). Results: A total of 133 adnexal masses were studied (benign: 66.2%, n:88; malignant: 33.8%, n:45) in a sample of women with mean age 56.5 ± 7.8 years. Malignant lesions were identified by SA in all cases. Borderline ovarian tumors (n:13) were not always detected by some US scores (SRRA: 76.9%, ADNEX model without and with CA125: 76.9% and 84.6%) nor were serous carcinoma (n:19) (SRRA: 89.5%), clear cell carcinoma (n:9) (SRRA: 66.7%) or endometrioid carcinoma (n:4) (ADNEX model without CA125: 75.0%). While most teratomas and serous cystadenomas have been correctly differentiated, other benign lesions were misclassified because of the presence of solid areas or papillae. Fibromas (n:13) were better identified by SA (23.1% malignancy), but worse with the other US scores (SRRA: 69.2%, ADNEX model without and with CA125: 84.6% and 69.2%, O-RADS: 53.8%). Cystoadenofibromas (n:10) were difficult to distinguish from malignant masses via all scores except SRRA (SA: 70.0%, SRRA: 20.0%, ADNEX model without and with CA125: 60.0% and 50.0%, O-RADS: 90.0%). Mucinous cystadenomas (n:12) were misdiagnosed as malignant in more than 15% of the cases in all US scores (SA: 33.3%, SRRA: 16.7%, ADNEX model without and with CA125: 16.7% and 16.7%, O-RADS:41.7%). Brenner tumors are also difficult to classify using all scores. Conclusion: Some malignant masses (borderline ovarian tumors, serous carcinoma, clear cell carcinoma, endometrioid carcinomas) are not always detected by US scores. Fibromas, cystoadenofibromas, some mucinous cystadenomas and Brenner tumors may present solid components/papillae that may induce confusion with malignant lesions. Most teratomas and serous cystadenomas are usually correctly classified.

## 1. Introduction

Various ultrasound (US) scores have been used in an attempt to differentiate malignant from benign adnexal masses, most of them based on the terms and definitions published in the year 2000 by the IOTA (International Ovarian Tumor Analysis) group [1]. The main characteristics taken into account in the US scores are the size, the number of locules, whether the internal wall is smooth or irregular (irregular would be considered if a cystic lesion has a papillary projection or the outer contour or a solid lesion is irregular), and the presence of septum (considered as a band of tissue that crosses a cystic mass from its inner surface to its contralateral side), a solid portion (described as echogenic tissue), papillary projections (defined as a solid projection in a cystic cavity ≥ 3 mm height), acoustic shadowing (seen as hypoechoic bands behind a structure) or ascites (if there is liquid outside the pouch of Douglas). The septa, solid parts or papillary projections can be examined with color Doppler to assess their degree of vascularization (score color 1: none; score color 2: low flow; score color 3: moderate; score color 4: intense). In addition, other clinical data include whether the US was performed in an oncology center, the age of the patient and the values of CA125.

It has already been demonstrated that a US study undertaken by an experienced sonographer (Subjective Assessment) is the best choice for classifying adnexal masses [2,3,4]. However, other US scores have been developed to help non-expert sonographers to classify adnexal masses such as IOTA Simple Rules Risk Assessment (2016), available as a digital version of the IOTA Simple Rules (2008) [5], which predicts the risk of malignancy in an online calculator [6]; the ADNEX model (Assessment of Different Neoplasia in the Adnexa) (2014) [7] with or without CA125; or the O-RADS system (Ovarian Adnexal Reporting and Data System) [8], introduced in 2020 by the American College of Radiology (ACR) [9] with an updated version in November 2022, which classifies adnexal masses into five categories according to lexicon descriptors and includes management options. However, despite all the efforts made, the diversity of presentation of the adnexal lesions continues to make it difficult to categorize them without mistake.

The aim of this study is to describe the ultrasound features of different adnexal masses and cancers that are difficult to be classified on the ultrasound and observe how they are analyzed by the Subjective Assessment of an experienced sonographer and the most commonly used US scores (Simple Rules Risk Assessment, ADNEX model and O-RADS).

## 2. Materials and Methods

This is a retrospective study of consecutive women who underwent surgery for adnexal masses from January 2021 to December 2022 in the Department of Gynecology of a tertiary-care university hospital in Madrid (Spain). Other adnexal masses with ambulatory ecoguided biopsy have been analysed elsewhere [10] and were not included in this sample.

The results of the global analysis of the sample have been previously published [11]. In this study, subgroups of patients with the most common adnexal pathology with non-easy classification according to US scores were studied.

Clinical information and ultrasound images/ultrasound medical reports were reviewed. We obtained approval from the Local Ethics Committee.

## 3. Criteria for Inclusion and Exclusion

The women included had a definitive histological study of the adnexal lesion, with a previous gynecological ultrasound (maximum 180 days before) (Figure 1). Images or ultrasound medical reports were stored in the hospital’s PACs or in the ultrasound software.

## 4. Methodology

Data collected from patients referred to their age, menopausal status, clinic (asymptomatic, digestive, bleeding, other), CA125 (IU/mL, considering normal values 0.0-35.0), surgical approach (laparoscopic/laparotomy, double adnexectomy with/without hysterectomy, conservative surgery—cystectomy/oophorectomy/unilateral adnexectomy and laterality—right, left, bilateral) and histopathology.

Results of CA125 were obtained by a Alinity i CA125 II Reagent Kit (Abbot, Chicago, IL, USA).

Transvaginal or transrectal images were taken from an RIC 5-9D 4–9 MHz endovaginal probe and an RAB6-D 2–8 MHz transabdominal probe Voluson E8 (GE Healthcare, Ultrasound, Milwaukee, WI, USA), Canon Aplio A and Xario 100 (Canon Medical Systems corporation, Tokyo, Japan).

The ultrasound features included largest size (mm), contour (regular/irregular), acoustic shadowing, presence of solid areas (mm) and their Doppler color (score color 1–4), septum (characteristics, score color 1–4), number of locules (none/1/2–9/ ≥ 10), presence of papillae (number, size, score color 1–4), and presence of ascites (no–mild/moderate–severe). Ultrasound examinations were performed by experienced/non-experienced gynecologists, all of them following the scanning system and the lexicon described by the IOTA group [1]. Images/clinical reports were automatically stored in the ultrasound software and in the Picture Archiving and Communication System (PACS)/electronical clinical history immediately after the ultrasound scan. The images/clinical reports were reviewed by two ultrasound gynecological experts with more than 15 years of ultrasound analysis experience who were blind to the pathologic results and other image findings (CT and/or MRI). In cases of disagreements, images were reviewed until a consensus was reached.

Ultrasound scores included Subjective Assessments, in which images/clinical reports were evaluated by two experienced ultrasound gynecologists with more than 15 years of experience (I.P.D. and L.A.M.) who described the adnexal mass according to IOTA criteria, with a final classification into benign or malignant masses. In cases of discrepancy, agreement ultrasound images were discussed to give a final conclusion. Both examiners were blind to definitive histopathological diagnoses and other clinical data.

IOTA Simple Rules Risk Assessment (%) is shown at https://homes.esat.kuleuven.be/~sistawww/biomed/ssrisk/ (last accessed 1 April 2023).

The ADNEX model with and without CA125 was obtained from https://www.iotagroup.org/sites/default/files/adnexmodel/IOTA%20–%20ADNEX%20model.html (last accessed 1 April 2023). The parameters describe the percentage of malignancy.

O-RADS was classified from 0 to 5 following direct access at https://www.acr.org/-/media/ACR/Files/RADS/O-RADS/O-RADS_US-Risk-Stratification-Table.pdf and https://www.acr.org/-/media/ACR/Files/RADS/O-RADS/US-v2022/O-RADS-v2022-Updates.pdf (last accessed on 1 April 2023).

Each of these scores takes into account some of the US characteristics for their classification, which are summarized in Table 1.

In all the scores we used a cut-off of 10% risk of malignancy to classify the mass. If the estimated risk of malignancy was equal to or higher than 10%, the mass was considered malignant. O-RADS stages 4 and 5 are considered malignant for classification purposes.

If the woman had more than one (bilateral) adnexal mass, the most complex mass—or if both were equal, the largest—was included.

Levels of CA125 were recorded on the same day as the ultrasound evaluation and were analyzed using an Alinity i CA125 II Reagent Kit (Abbot, Chicago, IL, USA).

A histological diagnosis was performed by a group of experts in gynecological pathology who classified the adnexal lesions in accordance with the guidelines of the World Health Organization [12,13,14]. Borderline tumors were considered malignant for classification purposes in this study.

## 5. Statistical Analysis

Excel software for Microsoft 365 MSO (64-bit version 2211) (Redmon, WA, USA) was used for both data recording and data processing, which also provided the basic statistics. For the analysis of the variables in this study, categorical variables were described by their absolute and relative frequencies, while continuous variables were represented by their mean and standard deviation. To assess changes in US score classification efficacy across different analyzed subgroups, Risk Ratios (RR) were calculated. These calculations were performed using the statistical software STATA 18.0.

## 6. Results

A total of 133 adnexal masses were studied; 66.2% were benign (n:88) and 33.8% malignant (n:45) in a sample of women with mean age 56.5 ± 7.8 years. Benign lesions included fibromas (n:13), cystoadenofibromas (n:10), mature cystic teratomas (n:29), mucinous cystadenomas (n:12), serous cystadenomas (n:19) and Brenner tumors (n:5). Malignant masses were borderline carcinomas (n:13), serous carcinomas (n:19), clear cell carcinomas (n:9) and endometrioid carcinomas (n:4).

The basal characteristics of patients, the ultrasound characteristics of the adnexal masses and their classification in the US scores are summarized in Table 2, Table 3 and Table 4.

### 6.1. Fibromas

The 13 cases of fibromas were found in women usually postmenopausal (10/13, 76.9%), more than 80% of which were over 50 years old. Almost half of them were asymptomatic (6/13, 46.1%) and some had digestive symptoms (3/13, 23.1%), and only one patient presented postmenopausal bleeding at the initial study. CA125 was increased in three patients (Table 2). Fibromas were described as big lesions with mean size 97.1 ± 42.3 mm (range: 50–188 mm) of solid predominance (84.6%, 11/13), except one multilocular (seven locules) and another with five locules and a solid part (Table 3) with thin and avascular septae. The contour was regular in all cases, with posterior shade in all but two cases (84.6%, 11/13). None had papillae. In nine cases, we found ascites (69.2%, 9/13); in four it was mild and in five it was moderate (Figure 2).

The SA classified benign lesions except in three cases (23.1%). SRRA was high (>10%) in nine cases (69.2%, 9/13) due to solid lesions with ascites, and some of them had a positive color map (n:1, 7.7%). The ADNEX without CA125 was >10% in all but two cases (84.6%, 11/13). ADNEX with CA125 was considered malignant in nine cases (69.2% 9/13), because of the solid appearance of fibromas. O-RADS in the 2018 version classified seven of them as malignant (53.8%, 7/13): five with intermediate risk (O-RADS 4) and two with high risk (O-RADS 5). In the new version 2022, the O-RADS category 3 includes a solid lesion with smooth contouring and shadowing, with any size and color score 2-3, which reclassified as benign three cases, thus increasing the sensitivity and specificity of the technique (Table 4). The SA classified fibromas as non-benign lesions in three cases (23.1%). SRRA misclassified nine cases (69.2%, 9/13) due to their solid appearance with ascites. Compared with the SA, the SRRA and ADNEX model scores increased by three the risk of misclassification (IC95% SRRA: 1.04, 8.63, *p*-value = 0.018; IC 95% ADNEX without CA125: 1.32,10.16, *p*-value = 0.002; IC 95% ADNEX with CA125: 1.04,8.63, *p*-value = 0.018) (Table 4).

### 6.2. Cystoadenofibromas

We had 10 cases of cystoadenofibromas, mainly in postmenopausal women (70.0% 7/10), with ages equal to or greater than 60 years. Most were asymptomatic (60.0%, 6/10), three had digestive clinic (30.0%, 3/30), and only one showed bleeding (Table 2). CA125 was elevated in only one patient (CA125: 51 IU/mL). Laparoscopy was performed in eight cases (six double anexectomy) and only two required laparotomy due to their large size (126 and 178 mm maximum diameter), even though most of them were smaller than 4 cm (60.0%, 6/10). Regarding the ultrasound characteristics (Table 3), all presented a regular contour, not always with posterior shading (absence of posterior shade n:4, 40.0%). Seven of them (70.0%, 7/10) presented a small solid area (6–17 mm), with absent/scarce vascularization. They mostly had one or more papillae (70.0%, 7/10) of variable sizes (4–17 mm), with absent/scarce vascularization. Only one case presented a thick but avascular septum. Most had one to two lobules (70.0%, 7/10), although in three cases the image showed a multilocular cyst with more than 10 locules (Figure 3). The subjective diagnosis of malignancy was high in most cases (70.0%, 7/10) (Table 4). IOTA SRRA predicted benignity in all but two cases (11.4% and 81.4%). The ADNEX Model without CA125 classified six as malignant (60.0%) and the ADNEX Model with CA125 achieved five false positives (50.0%). O-RADS classified all but one with a high score (O-RADS 4-5); most of them as uni/bilocular (O-RADS 4, 60.0%, 6/10). Two of them were unilocular lesions with four or more papillae, categorized as O-RADS 5. Although not statistically significant, SRRA was the best US score that could be used to predict the benignity of cystoadenofibromas (Table 4).

### 6.3. Mature Cystic Teratomas

In total, 29 cases of mature cystic teratomas were diagnosed in our series, most of them in premenopausal women aged between 14 and 48 years (82.7%, 24/29) (Table 2). CA125 values were only elevated in two cases (6.9%). A high laterality rate was found (25.6%, 8/29). They were either completely asymptomatic or had unspecific digestive symptoms. Surgery was usually laparoscopic (82.7%, 24/29), with cystectomy performed in 12 cases (41.4%). Regarding the ultrasound characteristics (Table 3) (Figure 4), the size could be very variable (31–220 mm), with a regular contour (93.1%, 27/29) and unilocularity (96.5%, 28/29), and some of them had some thin septum (n:3) with little vascularization. In only five cases, a solid portion was identified (17.2%), three of which had moderate vascularization. In one case, three papillae with moderate vascularization were found. Thick septae with little/no vascularization were described in two cases. Most presented acoustic shadowing (82.7%, 24/29) and no associated ascites. The SA, SRRA and ADNEX model without Ca125 (Table 4) classified as malignant only three cases, corresponding to lesions that had a moderately vascularized solid portion. The ADNEX model with CA125 failed in only one case (ADNEX model probability of malignancy 15.9%). Via O-RADS, most were classified correctly (O-RADS 2: 58.6%, 17/29; O-RADS 3: 31.0%, 9/29). Only the three cases mentioned above were included in the O-RADS 4-5. Both SA and the other scores were generally accurate in the classification of mature cystic teratomas as benign (Table 4).

### 6.4. Mucinous Cystadenoma

We found 12 cases of mucinous cystadenoma in pre (58.3%, 7/12) or postmenopausal women with a variable age range (28–69 years), usually asymptomatic (58.3%, 7/12) (Table 2). Lesions appeared unilaterally, especially in the right ovary (66.7%, 8/12). The type of surgery performed was usually oophorectomy (cystectomy in 1 case) by laparoscopy (83.3%, 10/12). In our sample, most of the lesions were large (10 of them larger than 8 cm) uni- or bilocular lesions (75%, 9/12), of regular contour and thin avascular septae, with posterior acoustic shadows (Table 3) (Figure 5). Only two cases presented a small avascular solid part (12–25 mm) and three of them also showed avascular papillae. None of the cases showed associated ascites. Regarding ultrasound classification (Table 4), SA classified as malignant 33.3% (4/12), and the SRRA and ADNEX models with or without CA125 classified 16.7%. Most lesions were O-RADS 3 (50.0%) and 4 (41.7%). O-RADS yielded the worst US score for predicting the benignity of mucinous cystadenomas (Table 4), although this was not statistically significant (Table 4).

### 6.5. Serous Cystadenoma

In our sample, serous cystadenomas (n:19) appeared in any age range (19–81 years), including in women who were pre- (52.6%, 10/19) and postmenopausal (47.3%, 9/19), most of them being asymptomatic (73.7%, 14/19) (Table 2). The CA125 values remained within normal ranges, except for one case. They usually appeared in the right adnexa (57.9%, 11/19), although 15.8% (3/19) were bilateral. Regarding the type of surgery, almost all were laparoscopies (89.5%, 17/19), with unilateral oophorectomy (n:6), cystectomy (n:4), salpinguectomy (n:1), unilateral anexectomy (n:1) and double anexectomy being performed. The two cases of laparotomy were larger than 14 cm, making laparoscopic access difficult. In the ultrasound, a wide variety of sizes (24–170 mm) were found, distributed homogeneously. Most of the cases were uni- or bilocular lesions, with fine avascular septae with acoustic shadows and regular contours (Table 3) (Figure 6). Only 10-15% of the masses were considered malignant in some US scores (SA, SRRA, ADNEX model). O-RADS considered all masses except three as benign (O-RADS 2: 31.6% 6/19, O-RADS 3: 52.6% 10/19, O-RADS 4: 10.5% 2/19, O-RADS 5: 5.3% 1/19) (Table 4). In the classification of serous cystadenomas as benign, both SA and the other scores were generally accurate (Table 4).

### 6.6. Brenner Tumor

We found five cases of Brenner tumor, all of them in postmenopausal women close to 60 years, who were generally asymptomatic (60%, 3/5) (Table 2). The CA125 value was only elevated in one patient (44 IU/mL). Regarding the location, most were in the right ovary, and there was one bilateral case. Hysterectomy with double anexectomy was performed in two cases by laparotomy, and double anexectomy was performed by laparoscopy in three cases. Regarding ultrasound features (Table 3), three of them were larger than 10 cm, with thin and avascular septa, and four cases were multiloculated formations (one of them with more than 10 locules), with posterior shade. In three of them (60.0%, 3/5), a solid component was found (one with moderate vascularization) (Figure 7). While the SRRA and ADNEX models with or without CA125 identified malignancy in 25.0% of the cases (Table 4), SA and O-RADS classified most of them as malignant (SA: 80.0% 4/5, O-RADS 4: 75.0% 3/5). Although there was no statistical significance, SA and O-RADS yielded the worst US classifications for predicting benignity in Brenner tumours (Table 4).

### 6.7. Borderline Carcinoma

In our series, we found 13 cases of borderline carcinoma of the ovary: 5 mucinous (BLM), 7 serous (BLS) and 1 mucinous and serous type case. The mean age of presentation varied in both groups in a wide age range, from 28 to 70 years (mean: 50.3 years ± 13.6). Half of the patients were not menopausal (53.8%, 7/13). Elevated CA125 was found in seven cases (five of them BLS) (Table 2). Initial laparoscopy with double anexectomy was performed in five cases (38.5%, four of them BLS). Six of the cases were diagnosed in the right ovary (46.1%) and the four bilateral cases were BLS. Half of them were asymptomatic (46.1%, 6/13) and the other half presented abdominal clinical features. Regarding ultrasound characteristics, the size was greater than 5 cm in all cases of BLS and 11 cm in the BLM. In almost all the masses (Table 3), a solid part (76.9%, 10/13) of very variable size (53.5 ± 60.5 mm, range: 12–210 mm) was identified, with moderate–intense vascularization in six of them (46.1%). We found eight cases with papillae (in one case, we found 10 papillae, 3 of which had moderate vascularization), which corresponded in all cases, except in one, with BLS. The two cases with more than 10 loculations were BLM (Figure 8). The detection of malignancy by SA was 100% (Table 4). The SRRA assessment gave results lower than 10% in three cases (23.1%), all of which were BLS. The ADNEX model with CA125 had high rates of detection. O-RADS classified all lesions as high-risk (O-RADS 4—n:6, 46.2%; O-RADS 5—n:7, 53.8%). When compared to the SA’s performance, the SRRA showed a seemingly reduced classification capacity of 0.77 (95% CI 0.57, 1.04), although statistical significance was not reached (Table 4).

### 6.8. Serous Carcinoma

Nineteen cases of serous carcinoma were diagnosed in women aged 40 and over (61.9 11.3 years, range 42–91 years), of whom only two were premenopausal (Table 2). CA125 was high in mostly all cases (89.5%,17/19) with variable ranges from 63 to 12059 IU/mL. In more than half of the cases the lesion was bilateral, affecting both ovaries (52.6%, 10/19). In the ultrasound study, there were lesions of variable sizes (from 44 to 160 mm) with irregular contours (73.7%, 14/19), and with solid predominance, all of which had a solid portion or papilla, and most of which had moderate/intense vascularization (68.4%, 13/19) (Figure 9). In five cases they presented with ascites (26.3%) (Table 3). In the SA (Table 4), all the masses were classified as malignant, while SRRA did not detect two cases (10.5%). Both the ADNEX model with and that without CA125 detected malignancy in all cases (range: 26.9–100%). O-RADS also classified all lesions as malignant (O-RADS 4: 42.1%, 8/19; O-RADS 5: 57.9%, 11/19) because of the presence of a solid component. Therefore, although not statistically significant, the SRRA is the only one that did not detect all cases of serous carcinoma (Table 4).

### 6.9. Clear Cell Carcinoma

Of the nine cases of clear cell carcinoma, all but one were diagnosed in women between 40 and 60 years, most of them postmenopausal (77.8%, 7/9) (Table 2). In three cases, the CA125 level was not elevated, while the rest showed very different values ranging 43.4–1411 IU/mL. The approach was laparotomic in most cases (77.8%, 7/9), and only one case of bilateral involvement was found. As for the ultrasound characteristics (Table 3), they were large lesions (>7 cm), and all of them had no posterior shadowing, with solid components of variable sizes (range: 24–155 mm), many of which had moderate/intense vascularization (77.8%, 7/9) (Figure 10). Only one case showed associated ascites. SA classified all cases as suspicious of malignancy (Table 4). The SSRA diagnosed 66.7% (6/9). The ADNEX models with/without CA125 classified 100% of lesions as malignant. According to O-RADS, all lesions showed a high risk of malignancy (O-RADS 4: 33.3%, 3/9; O-RADS 5: 66.7%, 6/9) as they presented as uni/multilocular lesions with a solid component. In summary, the SRRA is the only score that did not detect malignancy in all cases of clear cell carcinoma, although this was not statistically significant (Table 4).

### 6.10. Endometrioid Carcinoma

Four cases of endometrioid carcinoma were diagnosed in relatively young women (35–52 years), half of whom were premenopausal, and all had associated digestive symptoms (Table 2). The CA125 was elevated in all cases with variable values (CA 125: 74.9–1579.3 IU/mL). The most frequent ultrasound features found were (Table 3) large (>8 cm) uniloculated, predominantly cystic masses, with irregular contours (75.0%, 3/4); all of them had a solid part/papillae with moderate/intense vascularization, except in one case, and no posterior shadowing (75.0%, 3/4) (Figure 11). According to the SA, SRRA and ADNEX models with CA125, all lesions were classified as malignant (Table 4). The ADNEX model without CA125 indicated malignancy in 75.0% of lesions (3/4). O-RADS classified all lesions as showing a high risk of malignancy (O-RADS 4: 50.0%, 2/4; O-RADS 5: 50.0%, 2/4), corresponding to lesions with some type of solid component/papillae. In this case, the ADNEX model without CA125 was the only scoring method that did not detect malignancy in all cases (not statistically significant) (Table 4).

## 7. Discussion

Valentin et al. [15] studied a big group of adnexal masses (n:1066) that were difficult to classify by an experienced ultrasound examiner. They concluded that certain US characteristics were associated with unclassifiable masses, such as papillary projections, multilocularity (>10 locules) without solid components, and moderate vascularization, which could be found in some borderline tumors, cystadenomas or cystadenofibromas between other lesions.

The failure in diagnosing a malignant lesion and classifying it as benign may result in a delayed diagnosis, with the consequent progression and spread of the disease and a worsening prognosis. On the contrary, though, considering a benign lesion as malignant can increase the patient’s anxiety, lead to overly aggressive surgeries, and increase costs unnecessarily, among other factors.

In this study, we have analyzed the main ultrasound features of different adnexal masses and observed how different US scoring systems can classify them as benign or malignant.

Fibromas belong to the group of sex cord-stromal neoplasms [16]. In our series, fibromas were mostly asymptomatic or with mild digestive symptoms, as described in the literature [16,17]. They appear predominantly as big solid lesions with a regular contour and with posterior shading, as described by Chen et al. (26/27, 96.3%) [17]. Ascites, which were found to be associated in most cases, can be explained by transudation through the tumor surface or by irritation of the peritoneum, which would also increase CA125 in certain situations (in three patients in our series) [18]. The solid-like appearance of fibroids is sometimes confused with pedunculated myomas, which have a uterine-binding pedicle that should be discarded [19]. In most cases, the SA was able to infer a benign lesion, except in three cases: one because of its big size, another had several cystic areas, and another was twisted. Paladini et al. [16] described that almost 20% (12/68) of fibromas were misclassified by SA as malignant when they appeared with atypical morphology (irregular contour, cystic lesion, absence of acoustic shadowing). The ADNEX model without CA125 and SRRA classified most of them as malignant, as their appearance under ultrasound was similar to that of a solid tumor. Low levels of CA125 improved the rate of diagnosis in two cases. The new O-RADS classification (version November 2022) seems to be more adequate, as it includes in O-RADS Section 3 the category of a solid lesion with smooth contours and shadowing, with any size and color score 2–3, which permitted 9/13 fibromas. In comparison with the SA criteria, the SRRA and the ADNEX models without and with CA125 showed failure rates for the classification of fibroids as benign that were significantly higher than others (*p*-value = 0.018, 0.002 and 0.018, respectively).

The ultrasonographical description of cystadenofibromas can be difficult. Virgilio et al. [20] studied more than 200 serous cystadenofibromas and described up to 10 ultrasound patterns. According to our series, the most common ones are uni/multilocular solid cysts with papillary proyections or small solid components, such as one or more papillae (1–7 papillae) of variable sizes with absent/scarce vascularization. Other authors also found papillary projections in 56–69% of cystadenofibromas with no vascularization [21,22]. Valentin et al. [15] related the presence of papillary projections with masses of difficult classification that are present in most cystadenofibromas. This is the reason why most of the US scores (SA, ADNEX models with or without CA125 and O-RADS) usually classify them as malignant at a high percentage. IOTA SRRA was the best predictor, with only 20% of the cystoadenofibromas classified as malignant masses in comparison with SA, although this was not statistically significant (*p*-value: 0.070) probably because of the limited number in the sample.

Mature cystic teratomas can be described as uni/mulitlocuar cysts with mixed ecogenicity, acoustic shadowing and scarce vascularization [23]. The good agreement between preoperative ultrasound diagnosis and histopathological study was also confirmed by Heremans et al. in a big series (81.9%, 372/454) [23]. Associations between mature cystic teratomas and anti-N-methyl-D-aspartate receptor (anti-NMDAR) and encephalitis have been described previously [24]. In general, US scores manage to classify most teratomas correctly.

Most mucinous neoplasms are benign and typical of young women (aged 20–40 years) [25], as in our series (41.7% of woman under 50 years). The most common presentation was uni-/bilocular lesions, except one case with more than 10 locules, which are related to borderline mucinous tumors [26]. Only two cases presented a small avascular solid part, and three had avascular papillae. These facts are in accordance with the study of Pascual et al. [27], which found that benign mucinous cystadenomas can also have solid components and/or moderate/intense vascularization, making it difficult to differentiate them from borderline and malignant masses. IOTA SRRA and ADNEX models with or without CA125 yielded the best US scores for the prediction of benignity. SA classified as malignant more than 30% of the masses: three were large (larger diameter > 10 cm), and presented papillae (1–3), an irregular septum or a solid component with scarce vascularization. SRRA found only two cases that were positive (32.7% and 48.7%), coinciding with the cases described above. The ADNEX models with and without CA125 also classified as malignant two of those with one to three papillae. Of the five cases considered malignant in O-RADS (O-RADS 4), four of them were uni/multilocular formations with solid components, and another was without solid components but had a large size (>10 cm).

Regarding serous cystadenomas, they most commonly appear as unilocular cysts within walls, and with no septa or papillary projections [28], as in our sample. The subjective criterion (SA) and O-RADS were associated with benignity in all but in three cases. One of them was a 45-year-old woman with a multiloculated multicystic lesion (>10 locules) without posterior shadowing, and with a vascularized irregular septum (score color 3), which also contained a vascularized solid part of 20 mm (score color 3), which was highly suspicious in relation to malignancy (SRRA: 64%, ADNEX without CA125: 46.8%, ADNEX CA125: 47.1%, O-RADS: 5). There was another cystic lesion with two avascular papillae (larger than 13 mm in diameter), which was classified as O-RADS 4, without other ultrasound scores being suspicious for malignancy (SRRA: 1.1%, ADNEX without CA125: 3.6%; ADNEX CA125: 3.0%). The other lesion that was suspicious was a large multiloculated mass (five locules) (>13 cm), with a vascularized irregular septum (score 4) classified by O-RADS and SRRA as malignant (O-RADS 4, SRRA: 71.7%), while the ADNEX model predicted benignity for this case (ADNEX without CA125: 10%, ADNEX CA125: 6.8%). Therefore, in general, most US scores correctly classified serous cystadenomas, with some exceptions.

Brenner tumors are epithelial surface tumors of the ovary that predominantly appear in postmenopausal women [29]. Weinberger et al. [29] found that most of them can contain solid components (70%, 16/23) that are poorly vascularized. They also described multiple calcification characteristics of this type of tumor that we could not confirm in our series. SA classified most of them as malignant (4/5) due to the presence of solid component or multiloculation. Regarding SRRA and the ADNEX models without or with CA125, only one case that was a solid large lesion (>10 cm) with moderate vascularization was considered malignant. O-RADS classified three of the tumors as malignant (O-RADS 4), as these were multilocular with a solid part with low vascularization; in the new classification (version November 2022), these would be reclassified as O-RADS 3 (benign). Therefore, Brenner tumors are rare and difficult to classify using US scores.

Borderline ovarian tumors are characterized by cellular proliferation and nuclear atypia with no stromal invasion, but they can be related to microinvasion, intraepithelial carcinoma and non-invasive peritoneal implants [12]. The management of borderline tumors varies with respect to invasive tumors in terms of fertility preservation, which is of great importance, given that many of them appear in premenopausal women (53.8% in our series). Valentin et al. [15] found that borderline malignant tumors were very difficult to assess, and in a series of 55 cases, only 47% were correctly classified, and 24% were unclassifiable. In our sample, more than half of them had a papillae or a solid portion, which could be a clue in their characterization as malignant. Fagotti et al. [30] found that a solid portion of at least 14 mm, the presence of papillae with vascularization, or both, were the best parameters that could be used to differentiate an invasive ovarian tumor from a borderline or benign one with 100% sensitivity, and with 63, 63 and 80% of specificity, respectively, in premenopausal patients with unilocular–solid adnexal masses. Moro et al. [31] described the clinical and ultrasound features of different subclasses of malignant serous ovarian tumors, and confirmed that the borderline ovarian tumor could be described as unilocular–solid or multilocular–solid with solid papillary projection, and also there was an overlap in ultrasound appearance between borderline ovarian tumors and non-invasive low-grade serous ovarian carcinoma, both presenting as cysts with papillary projections. In addition, in 2015, Ludovisi et al. [32] described serous surface papillary borderline ovarian tumors (SSPBOTs), a rare morphologic variant of serous ovarian tumors that are typically confined to the ovarian surface, as irregular solid lesions surrounding normal ovarian parenchima. In our series, three cases in which the SRRA assessment was lower than 10% (all of them BLS) corresponded with single/bilocular cystic lesions containing one to two avascular papillae, two of which corresponded to cases that the ADNEX model without CA125 did not detect. The ADNEX model with CA125 also did not detect another case of a large (>11 cm) unilocular formation with no solid part or papillae, and without posterior shading, with non-elevated CA125. O-RADS classified all lesions as high-risk because all but one case (multiloculated > 10 lobes, no posterior shade) had a papilla or solid portion.

Serous carcinoma in our series appeared as uni/bilocular cysts with a solid part or papillae, most of which were highly vascularized. Suh-Burgmann et al. [33] also stated that the most frequent appearance was cystic solid mass (77.4%), with solid components in 97.4% of cases. In small tumors (<5 cm), a solid appearance was predominant, in accordance with Di Legge et al. [34], who demonstrated that, even if the lesion is very small, its ultrasound features can still help in the diagnosis when the ultrasound examination is performed by an ultrasonoghrapher with advanced skills. More recently, Bruno et al. [35] described a case of a Leydig tumour of 22 mm, confirming that even in cases with small dimensions, ultrasound examination could allow a correct diagnosis and permit the personalization of treatment. Ascites were not always associated with advanced stages (15% of moderate–severe of stages II–IV), similar to data published previously (18.0%) [33]. While the SA and ADNEX models with and without CA125 and O-RADS correctly assessed all cases, SRRA did not detect two masses corresponding to uni/bilocular lesions with solid portion/papillae (<4) and with low/moderate vascularization (score 2–3).

Women diagnosed with clear cell carcinoma are younger than the ones with serous carcinoma (in our series, 55.7 years vs. 61.9 years, respectively), especially if they develop from endometriomas. As described by Pozzati et al. [36], they can emerge as large unilateral masses (88.9% in our series) with solid components in all cases, which justifies the subjective classification as malignant masses. The SA and ADNEX models with and without CA125 and O-RADS identified malignancy in all of these. The SSRA did not diagnose three cases (SSRA % malignancy 0.5–5.7%) that showed with lesions that were predominantly cystic uni/bilocular, with regular contours and non-intense vascularization in the solid portion.

Endometrioid carcinomas, as described by Moro et al. [37], were found in our series; these are large (>8 cm) lesions, all of which have a solid part/papillae with moderate/intense vascularization. Most US scores (SA, SRRA, O-RADS) defined all the masses as malignant that corresponded to lesions with some type of solid component/papillae, except in one case, which the ADNEX model without CA125 classified as benign, corresponding to a cystic mass with posterior shadowing that contained a single avascular papilla with moderate ascites. In this case, when CA125 (584.4 IU/mL) was added, the risk was higher than 10%.

Multiple studies have demonstrated a high sensitivity of US scores, which may vary depending on the sample under study (SA: 87.8–93.9%, SRRA: 78.1–81.1%, ADNEX model with CA125: 95.1–94.3%; ADNEX without model CA125: 87.8–88.7%; O-RADS: 90.2–98.1%) [11,38], but lower specificity (SA: 69.1–80.2%, SRRA: 72.8–82.1%, ADNEX model with CA125: 74.1–82.8%, ADNEX model without CA125: 67.9–77.6%, O-RADS: 60.5–73.1%), which means that we must continue working to give the appropriate value to each of the ultrasound criteria associated with malignancy.

Throughout this study, adnexal lesions that are difficult to classify as benign or malignant were selected, and we analyzed their ultrasound characteristics and how the US scores contribute to their classification. Fortunately, most malignant masses have been included in the malignancy group. However, some benign ones still need to be refined. Fibroids were classified correctly by an expert sonographer and poorly by the rest of the US scores, as they present as solid lesions in many cases associated with ascites. Many cystoadenofibromas have malignancy characteristics (such as solid portion/papilla) that confuse both the subjective criterion and the other US scores. Some mucinous cystadenomas may also present solid components and/or moderate/intense vascularization that may also induce confusion with malignant lesions. Although it is difficult to conclude about Brenner tumors, given the small sample, they are also generally difficult to identify. Teratomas and serous cystadenomas are generally correctly classified, with some exceptions.

## 8. Strengths and Limitations

All the masses included in our study were diagnosed in the same hospital with the same classification criteria, both ultrasound and anatomopathological, which provides homogeneity to the sample. However, the small sample limits the generalization of the conclusions and the statistical analysis.

The retrospective nature of the study with image analysis may limit its reproducibility, although the availability of video captures of the images that were more difficult to classify has made it possible to increase the possible performance of studies similar to prospective studies.

To date, there are not many articles that analyze the results of applying US scores to each of the adnexal mass groups in order to investigate the ultrasound criteria that cause these scores to fail, by classifying the lesions incorrectly as benign or malignant. Unfortunately, given that the same histological types of lesions may have different ultrasound presentations, it is difficult to establish a single ultrasound reference image to replace histological diagnosis. Nevertheless, the US scores could be improved in terms of their classification criteria so as to increase their specificity values. More studies are needed to make the US scores more accurate in classification. A comparison of the US scores with other imaging techniques, such as MRI (O-RADS MRI), could also help in the categorization of adnexal masses.

## 9. Conclusions

Some malignant masses (borderline ovarian tumors, serous carcinoma, clear cell carcinoma, endometrioid carcinomas) are not always classified as malignant by some US scores. Fibroids, cystoadenofibromas, some mucinous cystadenomas and most Brenner tumors may present solid components/papillae and can be confused with malignant lesions. Most teratomas and serous cystadenomas are generally correctly classified.

## 10. Future Research Directions

Since there is no perfect ultrasound score that differentiates benign from malignant adnexal masses, their systematic ultrasound classification should be improved to try to reduce false positives and negatives.

## Figures and Tables

**Figure 1 diagnostics-13-02785-f001:**
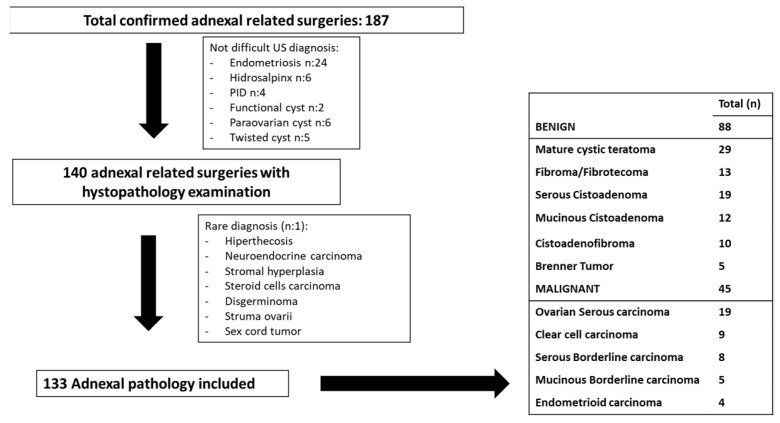
Flowchart of patients included in the present study and histological diagnoses of benign adnexal masses. US: ultrasound; PID: Pelvic Inflammatory Disease.

**Figure 2 diagnostics-13-02785-f002:**
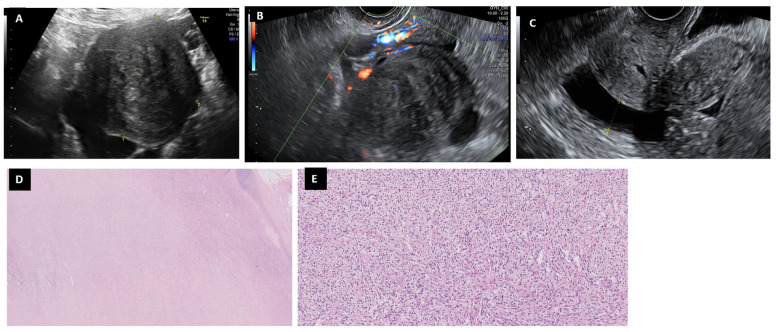
Fibroma in a 53-year-old woman. In the ultrasound is seen as a solid regular mass (**A**) with scarce Doppler color (score color 2) (**B**) with mild-moderate ascites (**C**). Interlacing bundles of spindle cell are characteristic of ovarian fibroma (H&E 2×) (**D**). Tumor cells are small and have a narrow ovoid nucleus (H&E 10×) (**E**).

**Figure 3 diagnostics-13-02785-f003:**
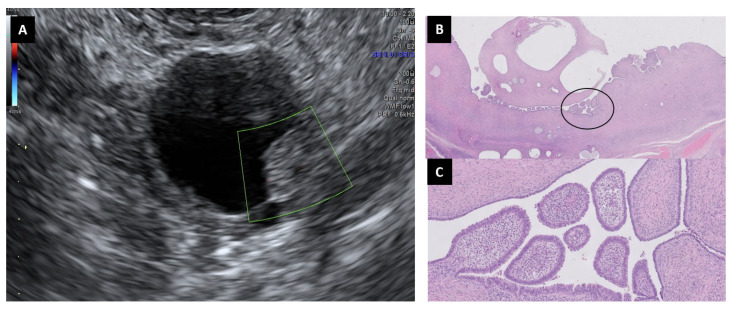
Cystoadenofibroma that includes a papillary avascular structure (**A**). Unilocular cysts can be seen with papillary projections to the lumen (H&E 2×) (**B**). Detail from the encircled area: papillary projections lined by a single layer of non-atypical tall, columnar, ciliated cells resembling normal tubal epithelium with stroma containing spindle fibroblasts (H&E 20×) (**C**).

**Figure 4 diagnostics-13-02785-f004:**
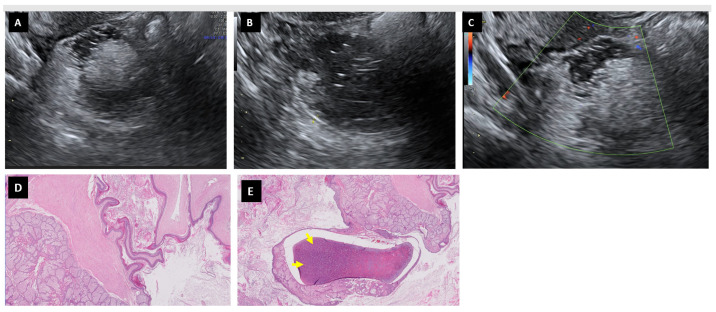
Mature cystic teratoma in a 25-year-old woman. Images (**A**,**B**) show a heterogeneous regular mass, sometimes difficult to differentiate from intestine with no Doppler color (score color 1) (**C**). Squamous epithelium and sebaceous glands form the wall of the teratoma, as well as abundant keratin debris, can be seen in the lumen (H&E 5×) (**D**). Cartilaginous tissue (arrows) was also present (H&E 5×) (**E**).

**Figure 5 diagnostics-13-02785-f005:**
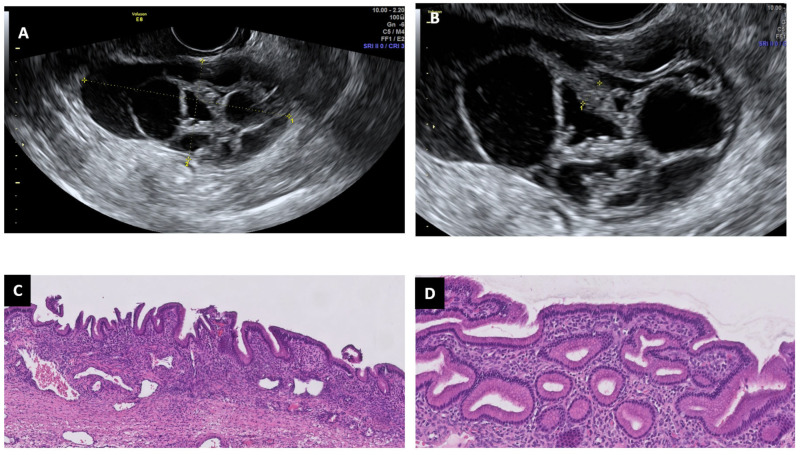
Ultrasound and histopathological images of a mucinous cystoadenoma: multilocular cyst with thick and irregular septums (**A**,**B**). This is a benign mucinous neoplasm composed of cysts and glands lined by Müllerian-type mucinous epithelium lacking architectural complexity or cytologic atypia (HE 5× and 10×) (**C**,**D**).

**Figure 6 diagnostics-13-02785-f006:**
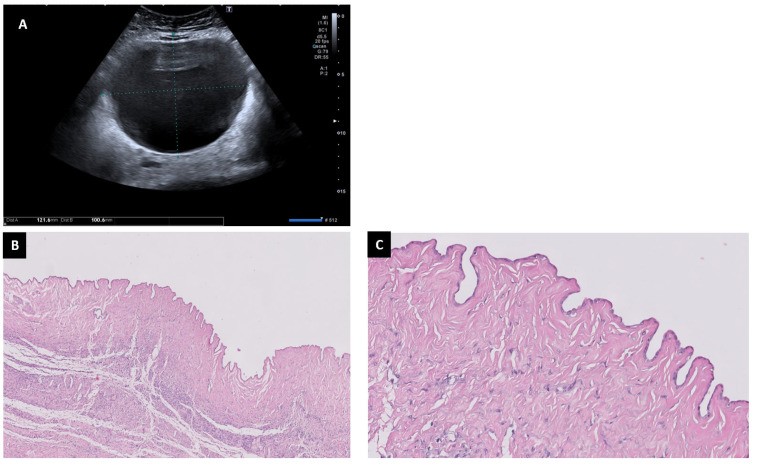
Serous cystoadenoma may appear as a regular cystic lesion without solid parts or papillae (**A**). The fibrous wall is covered by non-proliferative epithelia (H&E 2×) (**B**). Flat epithelial cells are shown without any atypia (H&E 20×) (**C**).

**Figure 7 diagnostics-13-02785-f007:**
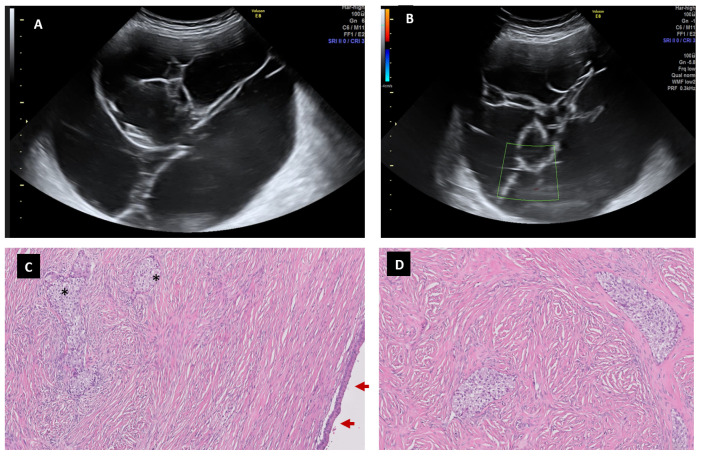
Seventy-eight-year-old woman with a Brenner tumor. A big multicystic lesion is observed (**A**) with thin avascular septum (**B**). Th eimage (**C**) shows a partially cystic mass, covered by mucinous epitelium (arrow). In the fibrotic wall, several nests of transitional cells can be seen (*) (H&E 5×). At higher magnification, we can see round to polygonal shapes with cell membranes distinct from the transitional nests, characteristic of Brenner tumors (H&E 10×) (**D**).

**Figure 8 diagnostics-13-02785-f008:**
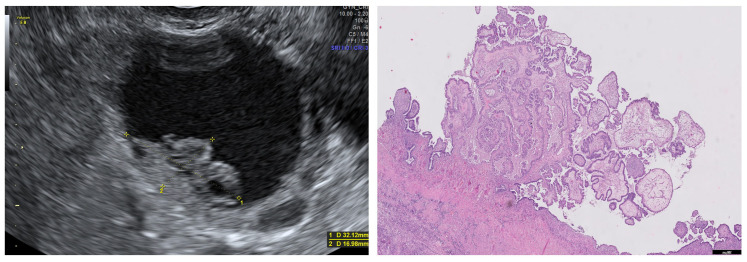
Ultrasound image of a borderline serous carcinoma with a nonvascularized papilla. Histologically, the papillae is composed of a fibrovascular stalk covered by proliferating serous epithelium (H&E 5×).

**Figure 9 diagnostics-13-02785-f009:**
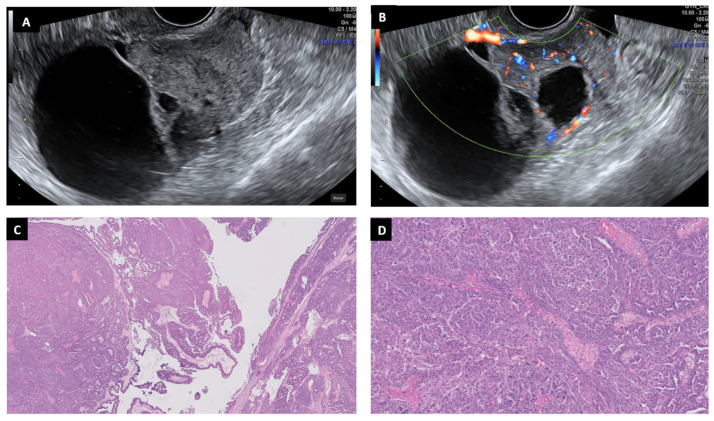
High-grade serous ovarian carcinoma in a 60-year-old woman: a solid cystic mass with moderate–intense Doppler color (score 3–4) is seen in ultrasound (**A**,**B**). Histopathology study shows a partially cystic solid neoplasm and papillary growth with hierachical branching (H&E 5×) (**C**). The luminal spaces are greatly narrowed by tumor cells creating slit-like spaces (H&E 10×) (**D**).

**Figure 10 diagnostics-13-02785-f010:**
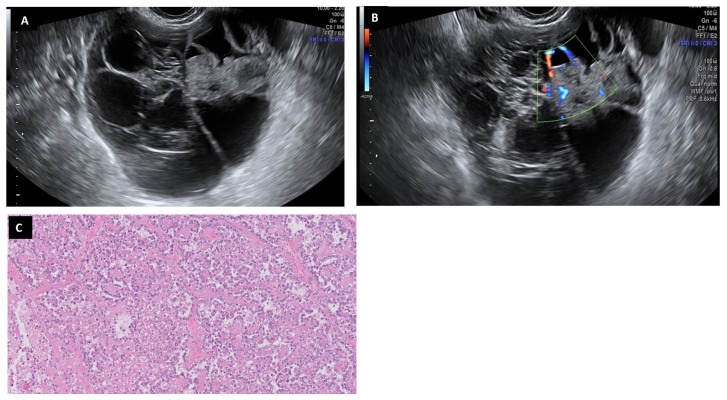
Irregular cystic mass with big solid vascularized part inside, corresponding to a clear cell carcinoma in a transvaginal ultrasound of a 43-year-old woman (**A**,**B**). Tubulo-cystic and papillary features (H&E 5×) are seen in the histopathology (**C**).

**Figure 11 diagnostics-13-02785-f011:**
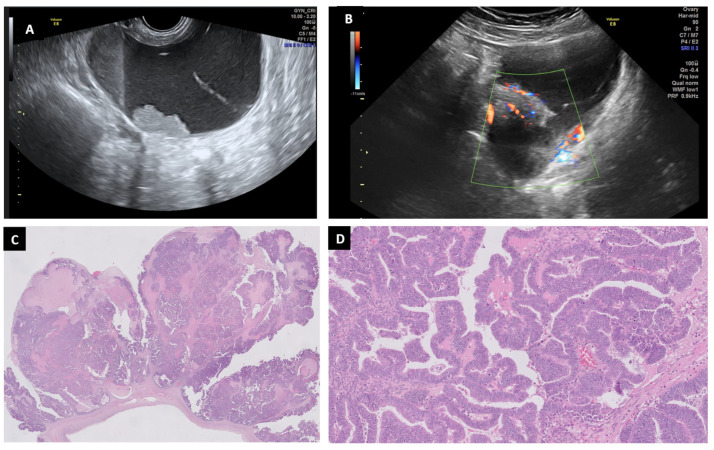
Ultrasound image of a large irregular cystic mass with irregular inner contours, and a papilla and solid part with moderate vascularization (score color 3) in a 51-year-old woman corresponding to an endometrioid carcinoma (**A**,**B**). Cystic mass with papillary projections (H&E 2×) (**C**). The tumor is composed of irregularly shaped endometrioid atypical glands (H&E 10×) (**D**).

**Table 1 diagnostics-13-02785-t001:** Summary of ultrasound features included in the different ultrasound scores evaluating adnexal masses.

	SRRA	ADNEX without CA125	ADNEX with CA125	O-RADS			
				2	3	4	5
Oncology center	Yes	Yes	Yes	No	No	No	No
Age	No	Yes	Yes	No	No	No	No
Size	≥100 mm M	Yes	Yes	Yes	Yes	Yes	No
Uni/Multilocular	Yes	> or <10 locules	> or <10 locules	Yes	Yes	Yes	Yes
Smooth/Irregular	Yes	No	No	Yes	Yes	Yes	Yes
Solid components	Yes	Yes	Yes	No	Yes	Yes	Yes
Size solid components	<7 mm B	Yes	Yes	No	No	No	No
Papillae	≥4 M	0/1/2/3/ > 3	0/1/2/3/ > 3	No	No	<4	≥4
Septum	No	No	No	Yes	Yes	No	No
Acoustic shadows	Yes	Yes	Yes	No	Yes *	Yes	Yes *
Blood flow (color score)	1:B 4:M	No	No	No	Yes	Yes	Yes
Ascites	M	Yes	Yes	No	No	No	Yes
CA125	No	No	Yes	No	No	No	No
Other features	-	-	-	Typical lesions	-	-	-

IOTA SRRA: International Ovarian Tumor Analysis Simple Rules Risk Assessment; ADNEX model: Assessment of Different Neoplasias in the Adnexa; O-RADS: Ovarian Adnexal Reporting and Data System; B: bening; M: malignant. * Only included in O-RADS US v2022.

**Table 2 diagnostics-13-02785-t002:** Basal characteristics of patients with benign or malignant adnexal masses.

	Age, Years (Mean, SD)	Postmenopausal (n, %)	CA125, IU/mL (Mean, SD)
Benign			
Fibroma (n:13)	62.7 ± 10.9	10 (76.9%)	93.2 ± 222.5
Cystoadenofibroma (n:10)	57.3 ± 9.5	7 (70.0%)	18.7 ± 14.0
Mature cystic teratoma (n:12)	36.5 ± 13.8	5 (17.2%)	21.6 ± 14.9
Mucinous cystoadenoma (n:12)	50.9 ± 11.7	5 (41.7%)	26.7 ± 21.1
Serous cystoadenoma (n:19)	51.2 ± 19.1	9 (47.3%)	15.1 ± 9.4
Brenner tumor (n:5)	66.4 ± 7.7	5 (100.0%)	29.9 ± 13.3
Malignant			
Borderline (n:13)	50.3 ± 13.6	6 (46.2%)	49.5 ± 7.7
Serous carcinoma (n:19)	61.9 ± 11.3	17 (89.5%)	1661.4 ± 3414.7
Clear cell carcinoma (n:9)	55.7 ± 10.7	7 (77.8%)	216.2 ± 451.3
Endometrioid carcinoma (n:4)	44.3 ± 8.5	2 (50.0%)	755.0 ± 625.0

**Table 3 diagnostics-13-02785-t003:** Ultrasound characteristics of benign or malignant adnexal masses. ↑ Doppler: Vascularization score color 3-4.

	Maximum Size (mm) (Mean, SD)	Solid Part			Papillae (n, %)	Acoustic Shadows (n, %)	Ascites (n, %)
		(n, %)	Size (mm) Mean (SD) Min.–Max.	↑ Doppler			
Benign							
Fibroma (n:13)	97.1 ± 42.3	11 (84.6%)	76.8 ± 23.2 (44–112)	1 (7.7%)	0 (0.0%)	11 (84.6%)	9 (69.2%)
Cystoadenofibroma (n:10)	60.0 ± 51.1	7 (70.0%)	12.1 ± 3.9 (6–17)	0 (0.0%)	0 (0.0%)	6 (60.0%)	0 (0.0%)
Mature cystic teratoma (n:12)	82.3 ± 40.1	5 (17.2%)	37.6 ± 18.1 (17–66)	3 (10.3%)	1 (3.4%)	24 (82.7%)	0 (0.0%)
Mucinous cystoadenoma (n:12)	121.5 ± 78.8	2 (16.7%)	18.5 ± 9.2 (12–25)	0 (0.0%)	3 (25.0%)	12 (100.0%)	0 (0.0%)
Serous cystoadenoma (n:19)	85.0 ± 39.6	2 (10.5%)	16.5 ± 4.9 (13–20)	1 (5.3%)	1 (5.2%)	16 (84.2%)	0 (0.0%)
Brenner tumor (n:5)	108.4 ± 43.1	3 (60.0%)	75.3 ± 34.5 (32–140)	1 (20.0%)	0 (0.0%)	4 (80.0%)	0 (0.0%)
Malignant							
Borderline (n:13)	50.3 ± 13.6	6 (46.2%)	53.5 ± 60.5 (12–210)	6 (46.1%)	8 (61.5%)	5 (38.5%)	2 (15.4%)
Serous carcinoma (n:19)	61.9 ± 11.3	17 (89.5%)	53.6 ± 23.9 (17–98)	13 (68.4%)	8 (42.1%)	4 (21.0%)	5 (26.3%)
Clear cell carcinoma (n:9)	55.7 ± 10.7	7 (77.8%)	53.0 ± 39.9 (24–155)	7 (77.7%)	2 (22.2%)	0 (0.0%)	1 (11.1%)
Endometrioid carcinoma (n:4)	44.3 ± 8.5	2 (50.0%)	47.0 ± 6.0 (41–53)	3 (75.0%)	3 (75.0%)	1 (25.0%)	1 (25.0%)

**Table 4 diagnostics-13-02785-t004:** Adnexal lesions classified as malignant according to the Subjective Assessment of the ultrasonographer, SRRA (Simple Rules Risk Assessment), ADNEX model without or with CA125 and O-RADS. The comparisons have been made with reference to the Subjective Assessment of the ultrasonographer.

	Subjective Assessment Malignancy		SRRA % Malignancy		ADNEX without CA125 Malignancy		ADNEX with CA125 Malignancy		O-RADS	
	No. Malignancy (%)	RR [95% CI] *p*-Value	No. Malignancy (%)	RR [95% CI] *p*-Value	No. Malignancy (%)	RR [95% CI] *p*-Value	No. Malignancy (%)	RR [95% CI] *p*-Value	No. Malignancy (%)	RR [95% CI] *p*-Value
Benign										
Fibroma (n:13)	3 (23.1%)	[reference]	9 (69.2%)	3 [1.04, 8.63] 0.018	11 (84.6%)	3.66 [1.32, 10.16] 0.002	9 (69.2%)	3 [1.04, 8.63] 0.018	7 (53.9%)	2.33 [0.77, 7.10] 0.107
Cystoadenofibroma (n:10)	7 (70.0%)	[reference]	2 (20.0%)	0.28 [0.08, 1.05] 0.070	6 (60.0%)	0.86 [0.45, 1.64] 1.000	5 (50.0%)	0.72 [0.34, 1.50] 0.650	9 (90.0%)	1.29 [0.82, 2.03] 0.582
Mature cystic teratoma (n:12)	3 (10.3%)	[reference]	2 (6.9%)	0.67 [0.12, 3.70] 1.000	3 (10.3%)	1 [0.22, 4.55] 1.000	1 (3.4%)	0.33 [0.04, 3.02] 0.611	3 (10.3%)	1 [0.22, 4.55] 1.000
Mucinous cystoadenoma (n:12)	4 (33.3%)	[reference]	2 (16.7%)	0.5 [0.11, 2.23] 0.640	2 (16.7%)	0.5 [0.11, 2.23] 0.640	2 (16.7%)	0.5 [0.11, 2.23] 0.640	5 (41.7%)	1.25 [0.44, 3.55] 1.000
Serous cystoadenoma (n:19)	3 (15.8%)	[reference]	3 (15.8%)	1 [0.23, 4.34] 1.000	2 (10.5%)	0.67 [0.13, 3.55] 1.000	2 (10.5%)	0.67 [0.13, 3.55] 1.000	3 (15.8%)	1 [0.23, 4.34] 1.000
Brenner tumor (n:5)	4 (80.0%)	[reference]	1 (25.0%)	0.25 [0.04, 1.52] 0.206	1 (25.0%)	0.25 [0.04, 1.52] 0.206	1 (25.0%)	0.25 [0.04, 1.52] 0.206	3 (75.0%)	0.75 [0.32, 1.74] 1.000
Malignant										
Borderline (n:13)	13 (100%)	[reference]	10 (76.9%)	0.77 [0.57, 1.04] 0.220	10 (76.9%)	0.77 [0.57, 1.04] 0.220	11 (84.6%)	0.84 [0.67, 1.07] 0.480	13 (100%)	1
Serous carcinoma (n:19)	19 (100%)	[reference]	17 (89.5%)	0.89 [0.77, 1.04] 0.486	19 (100%)	1	19 (100%)	1	19 (100%)	1
Clear cell carcinoma (n:9)	9 (100%)	[reference]	6 (66.7%)	0.66 [0.42, 1.06] 0.206	9 (100%)	1	9 (100%)	1	9 (100%)	1
Endometrioid carcinoma (n:4)	4 (100%)	[reference]	4 (100%)	1	3 (75.0%)	0.75 [0.46, 1.32] 1.000	4 (100%)	1	4 (100%)	1

## Data Availability

Data are available upon reasonable request.

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
