# Peer review of "Application of Ultrasound Scores (Subjective Assessment, Simple Rules Risk Assessment, ADNEX Model, O-RADS) to Adnexal Masses of Difficult Classification"

_diagnostics, 2023, doi:10.3390/diagnostics13172785_

Round 1
Reviewer 1 Report
Thank you for requesting to provide a review of this article about the application of ultrasound scores to adnexal masses that are difficult to be classified.
The main purpose of the analysis was to investigate the ultrasound features of some adnexal masses and to classify them in accordance with the most frequently used ultrasonographic scores. The main question adressed in the research was if it is possible to describe the ultrasound features of adnexal masses difficult to be classified by the ultrasound and to observe how they are being classified by the most used ultrasonographic scores.
The study is a retrospective analysis, in which 133 adnexal masses were studied for a period of time between January 2021 and December 2022. The topic is original and relevant in the field and brings usefull knowledge regarding the subject. A comprehensive search strategy was used. The review methodology was comprehensive with screening and data extraction. When it comes to the methodology used, no specific improvements should be considered from my point of view.
The conclusions are consistent with the evidence and the arguments presented, and they adress properly to the main question which conducted the analysis.
The references are appropriate and well suited for this kind of study.
The tables and figures used in the article are very precise and explicit and the information written there is very easy to be followed, so no other improvements are required from my point of view.
Regarding the structure and accuracy of the phrases, the manuscript has well structured information, with supported evidence and well structured phrases.
The manuscript is original and well defined. The results provide an advance in current knowledge. The results are being interpreted appropriately and are significant, as well as the conclusions.
The study is correctly designed and the analysis is being performed at high standards, so the data are robust enough to draw the conclusion. Surely the paper will attract a wide readership.
To conclude, the article is written in a proper way and brings useful information regarding the subject. However I have some issues to add in the lines below, but the article is suitable enough to warrant publication after the corrections are made:
Line 26: to the most, not „to most”
Line 29: the masses, not „masses”
Line 29: according to the, not „according to”
Line 58: „,” after „masses”
Line 58: based on, not „based in”
Line 70: includes, not „include”
Line 84: the presentation diversity, not „the diversity of presentation”
Line 108-109: of adnexal masses and cancers that are difficult to be classified on the ultrasound and observe..., not „of adnexal masses of difficult US classification and cancers and observe...”
Line 113: women who, not „women that”
Line 114: adnexal masses, not „adnexal surgery”
Line 120: were analyzed, not „is analyzed”
Line 124: The women, not „Women”
Line 142: „,” after „gynecologists”
Line 166: all the scores, not „all scores”
Line 170: the largest was included, not „the largest”
Line 171: on the same day with, not „on the same day of”
Line 285: most of the, not „most lesions”
Line 370: there were, not „they were”
Line 446: had studied a big group, not „studied in a big group”
Line 446-447: that was difficult to be classified, not „which could be difficult to classify”
Line 452: The failure in diagnosing, not „Failure to diagnose”
Line 565: diagnosed with, not „diagnosed of”
Line 607: A retrospective, not „Retrospective”
Line 26: to the most, not „to most”
Line 29: the masses, not „masses”
Line 29: according to the, not „according to”
Line 58: „,” after „masses”
Line 58: based on, not „based in”
Line 70: includes, not „include”
Line 84: the presentation diversity, not „the diversity of presentation”
Line 108-109: of adnexal masses and cancers that are difficult to be classified on the ultrasound and observe..., not „of adnexal masses of difficult US classification and cancers and observe...”
Line 113: women who, not „women that”
Line 114: adnexal masses, not „adnexal surgery”
Line 120: were analyzed, not „is analyzed”
Line 124: The women, not „Women”
Line 142: „,” after „gynecologists”
Line 166: all the scores, not „all scores”
Line 170: the largest was included, not „the largest”
Line 171: on the same day with, not „on the same day of”
Line 285: most of the, not „most lesions”
Line 370: there were, not „they were”
Line 446: had studied a big group, not „studied in a big group”
Line 446-447: that was difficult to be classified, not „which could be difficult to classify”
Line 452: The failure in diagnosing, not „Failure to diagnose”
Line 565: diagnosed with, not „diagnosed of”
Line 607: A retrospective, not „Retrospective”
Author Response
Thank you very much for your comments, they have been very useful to us, especially all the grammatical corrections, which have improved the quality of the scientific expression.
These contributions are marked in red in the file.
Thank you once again.

Reviewer 2 Report
This manuscript perfectly describes an investigation of the sonographic evaluation of adnexal masses to determine their malignant potential. The specific available diagnostic tools which were looked at are clinically pertinent, and which have also been investigated by others (and published). Using sonography to determine the malignant potential of identified adnexal masses is certainly important to address the difficulty that exists in the timely diagnosis of ovarian malignancy. It is especially laudable to correlate sonographic findings with what is ultimately found at an operational intervention. While efforts were made to compare diagnostic modalities for neoplasms that were found, the conclusion that SA yielded the best diagnostic results (compared to SRA, ADNEX, and O-RADS) for diagnosing malignancy, may be questioned (Table 4). If O-RADS category 4 or 5 (>10-50% ROM) would likely prompt a surgical exploration (so 0% of malignancies would be missed), the simple app may be most clinically useful when identifying sonographic descriptors for many clinicians. Perhaps the practicality of applying this tool should be mentioned in the comparison depicted.
The manuscript is well-structured, and illustrates IOTA-derived recommendations quite well.
Author Response
Thank you very much for your comments, they have been very useful to us.
QUESTION 1: While efforts were made to compare diagnostic modalities for neoplasms that were found, the conclusion that SA yielded the best diagnostic results (compared to SRA, ADNEX, and O-RADS) for diagnosing malignancy, may be questioned (Table 4).
ANSWER: The subjective assessment of the sonographer has the greatest diagnostic validity in the characterization of the adnexal masses, and it’s completely dependent on the experience and training of the examiner (1-3). In an attempt to make the classification of these lesions more objective and replace the expert’s experience as much as possible, different ultrasound scores have been proposed such as the US scores that we have analysed (SRA, ADNEX, and O-RADS). In a previous study of our group we concluded that IOTA SR, IOTA SRRA, and ADNEX models with or without CA125 and O-RADS can help in the differentiation of benign and malignant masses, and their performance is similar to the subjective assessment of an experienced sonographer (4). However, there are certain adnexal lesions that are difficult to classify as benign or malignant even for an ultrasound expert, or with the help of ultrasound scores, and these are the lesions that we have analysed in this paper.
- Viora, E.; Piovano, E.; Poma, C.B.; Cotrino, I.; Castiglione, A.; Cavallero, C.; Sciarrone, A.; Bastonero, S.; Iskra, L.; Zola, P. The ADNEX model to triage adnexal masses: An external validation study and comparison with the IOTA two-step strategy and subjective assessment by an experienced ultrasound operator. Eur. J. Obstet. Gynecol. Reprod. Biol. 2020, 247, 207–211.
- Jeong, S.Y.; Park, B.K.; Lee, Y.Y.; Kim, T.J. Validation of IOTA-ADNEX Model in Discriminating Characteristics of Adnexal Masses: A Comparison with Subjective Assessment. J. Clin. Med. 2020, 9, 2010.
- Tavoraite, I.; Kronlachner, L.; Opolskien ˙ e, G.; Bartkeviˇcien ˙ e, D. Ultrasound Assessment of Adnexal Pathology: Standardized ˙ Methods and Different Levels of Experience. Medicina 2021, 57, 708.
- Pelayo, M.; Pelayo-Delgado, I.; Sancho-Sauco, J.; Sanchez-Zurdo, J.; Abarca-Martinez, L.; Corraliza-Galán, V.; Martin-Gromaz, C.; Pablos-Antona, M.J.; Zurita-Calvo, J.; Alcázar, J.L. Comparison of Ultrasound Scores in Differentiating between Benign and Malignant Adnexal Masses. Diagnostics 2023, 13, 1307
QUESTION 2: If O-RADS category 4 or 5 (>10-50% ROM) would likely prompt a surgical exploration (so 0% of malignancies would be missed), the simple app may be most clinically useful when identifying sonographic descriptors for many clinicians. Perhaps the practicality of applying this tool should be mentioned in the comparison depicted.
ANSWER: In 2020 was published the Ovarian Adnexal Reporting and Data System (O-RADS) [1]. It classifies adnexal masses according to 6 categories and includes probabilities or risk of malignancy and guidelines for management according to the risk category. O-RADS 0 means that the evaluation is incomplete; O-RADS 1 is used for normal ovaries or a physiological cyst with 0% probability of malignancy; O-RADS 2 (<1% malignancy) is set for an almost certainly benign lesion; O-RADS 3 is used for lesions with low risk of malignancy (1–9%); O-RADS 4 indicates lesions with an intermediate risk of malignancy (10–49%), whereas O-RADS 5 is associated with a high risk of malignancy (≥50%). We agree that from the practical point of view O-RADS 4 and 5 are lesions that are recommended to be surgically studied even though they are intermediate (O-RADS 4) /high (O-RADS 5) risk but it can be a clinically different referring to the type of surgery, the type of surgical incision, the time to programme surgery,… In any case, the aim of the study is not to validate the usefulness of ultrasound scores, but to apply existing scores to lesions that are difficult to classify.
- Andreotti, R.F.; Timmerman, D.; Strachowski, L.M.; Froyman, W.; Benacerraf, B.R.; Bennett, G.L.; Bourne, T.; Brown, D.L.; Coleman, B.G.; Frates, M.C.; et al. O-RADS US Risk Stratification and Management System: A Consensus Guideline from the ACR Ovarian-Adnexal Reporting and Data System Committee. Radiology 2020, 294, 168–185.)
Reviewer 3 Report
Dear Editors,
thank you for the opportunity to review this manuscript. This is a single-center retrospective study from a university site that includes 133 adnexal masses between January 2021 to December 2022 that also underwent surgery. Overall of the 133 lesions were benign: 66.2%, n: 88 and malignant: 33.8%, n: 45.
Different entities were included and compared including: benign lesions included fibromas (n:13), cystoadenofibromas (n:10), mature cystic teratomas (n:29), mucinous cystadenomas (n: 12), serous cystadenomas (n:19) and Brenner tumors (n:5). Malignant masses were borderline carcinomas (n:13), serous carcinomas (n:19), clear cell carcinomas (n:9) and endomtrioid carcinomas (n:4).
All women included had a definitive histological study of the adnexal lesion, within a maximum 180 days to the examination. Clinical information were evalueted including age, menopausal status, CA125, surgical approach (laparoscopic/laparotomy, double adnexectomy with/without hysterectomy, conservative surgery) and histopathology.
Unfortunately, the results section is mainly descriptive. The illustrations in the results section are good and overall the work is valuable but needs major revision and improvement.
Some changes should be made:
-the results in the abstract are confusing and too long
-please add mean age of the patients already in the results part of the abstract
-Please add number of ethical approval
-Please add a flow-chart for patient recuritment to make it more transparent
-Please add a dedicated section for statistical analysis
-Table1 is included in the introduction, this should be changed into material and methods. Perhaps the table could be improved with figures or images
-I ask that the evaluated score not only be placed via a link but also be presented graphically or in an overview table in a better comparison.
-Please add a short table for the evaluated clinical information (age, laboratory values, ...) and for evaluated ultrasound features.
-Please add a figure or flow-chart of the different benign and malign entities that were included and add the individual numbers of patients
-Unfortunately, the results section is mainly descriptive. A multivariate analysis is essential. What are the different p-values?
-I don' t understand the chart of figure 11. Please explain it
-The results part must be structured: first descriptive information, then all benign and then all malignant lesions. Then the comparison with p-values.
-Please remove the red colour from the discussion section.
Author Response
REVIEWER 3: Answers in the paper are marked in blue
Thank you very much for your comments, they have been very useful to us.
- The results in the abstract are confusing and too long: Changed in paper (Tables 2-4)
- Please add mean age of the patients already in the results part of the abstract.
ANSWER: Added in paper in abstract and results: A total of 133 adnexal masses were studied (benign: 66.2%, n: 88; malignant: 33.8%, n: 45) in a sample of women with mean age 56.5±7.8 years.
- Please add number of ethical approval:
ANSWER: At the end of the paper it´s already included: “Institutional Review Board Statement: The study was conducted in accordance with the Declaration of Helsinki and approved by the Institutional Review Board of Hospital Universitario Ramón y Cajal (protocol code ECOSCORE1, approved on 20 December 2022)”.
4.Please add a flow-chart for patient recuritment to make it more transparent: Changed in paper (Figure 1)
- Please add a dedicated section for statistical analysis:
ANSWER: We appreciate your feedback. In response, we have reorganized the paper by separating the Methods section and introducing a new section dedicated to Statistical Analysis.
- Table1 is included in the introduction, this should be changed into material and methods. Perhaps the table could be improved with figures or images: Changed in paper
- I ask that the evaluated score not only be placed via a link but also be presented graphically or in an overview table in a better comparison.
ANSWER: Table 1 summarizes all the items included in the scores. As most of them are online calculators it´s difficult to include an image of the process.
- Please add a short table for the evaluated clinical information (age, laboratory values, ...) and for evaluated ultrasound features.
ANSWER: Added in paper. Table 2
- Please add a figure or flow-chart of the different benign and malign entities that were included and add the individual numbers of patients
ANSWER: Changed in paper: added in Figure 1.
- Unfortunately, the results section is mainly descriptive. A multivariate analysis is essential. What are the different p-values?
ANSWER: We appreciate the editor’s comment, and we agree that it was necessary to include a measure of differences between the US scores to strengthen our findings and conclusions. We included in the main text the table 4, which described the failures at diagnosis with the different scores and the risk ratio and p-value for each of them. Our objective was to describe how adnexal masses are classified by the most used US scores, but not to assess the effect of this classification on any defined outcome. We believe that multivariate analysis is not well suited to our objective. We have already published the comparison of Ultrasound Scores in Differentiating between Benign and Malignant Adnexal Masses (Pelayo M, Pelayo-Delgado I, Sancho-Sauco J, Sanchez-Zurdo J, Abarca-Martinez L, Corraliza-Galán V, Martin-Gromaz C, Pablos-Antona MJ, Zurita-Calvo J, Alcázar JL. Comparison of Ultrasound Scores in Differentiating between Benign and Malignant Adnexal Masses. Diagnostics (Basel). 2023 Mar 30;13(7):1307. doi: 10.3390/diagnostics13071307. PMID: 37046525; PMCID: PMC10093240) and also the value of the different parameters of the ultrasound scores in the classification of adnexal masses in benign or malignant (Pelayo M, Sancho-Sauco J, Sanchez-Zurdo J, Abarca-Martinez L, Borrero-Gonzalez C, Sainz-Bueno JA, Alcazar JL, Pelayo-Delgado I. Ultrasound Features and Ultrasound Scores in the Differentiation between Benign and Malignant Adnexal Masses. Diagnostics (Basel). 2023 Jun 23;13(13):2152. doi: 10.3390/diagnostics13132152. PMID: 37443546; PMCID: PMC10341072).
- I don' t understand the chart of figure 11. Please explain it
ANSWER: We had removed it. Table 4 summarizes better the information.
- The results part must be structured: first descriptive information, then all benign and then all malignant lesions. Then the comparison with p-values.
ANSWER: We appreciate the editor’s comment and we re-ordened the Results sections. However, as mentioned above, we have already published the comparison of ultrasound scores in differentiating between benign and malignant adnexal masses (see comment 10). Our aim was to describe how adnexal masses are classified by the most used US scores in each subgroup. We have slightly rewritten the main objective to make it clearer that our aim is to observe the classification trend within each of the subgroups of interest:
“… The aim of this study is to describe the ultrasound features of different adnexal masses and cancers that are difficult to be classified on the ultrasound and observe how they are classified by the most used US scores (Subjective Assessment, Simple Rules Risk Assessment, ADNEX model and O-RADS).”
- Please remove the red colour from the discussion section.
ANSWER: Sorry, we don´t see red colour in the version received. It may correspond to previous editor´s corrections.

Reviewer 4 Report
This is a good study with the limitations of being retrospective and including a relatively small number of patients. In addition, the large number of parameters studied mekes statistical evaluation of the results difficult. Usually no treatment decision is made based on the ultrasound examination alone. All or most of the women would also have undergone CT and/or MRI. It would be interesting to compare the results of ultrasound study with those of other imaging methods. It is of course clear thath the purpose of the article is limited to the ultrasound study but it would be very helpful in the discussion to have a comment on the comparison with the other methods.
The word projection is misspelled in some places and needs to be corrected.
Author Response
REVIEWER 4
This is a good study with the limitations of being retrospective and including a relatively small number of patients. In addition, the large number of parameters studied makes statistical evaluation of the results difficult.
ANSWER: We have simplified results in paper, I hope they are easier to analyse
Usually no treatment decision is made based on the ultrasound examination alone. All or most of the women would also have undergone CT and/or MRI. It would be interesting to compare the results of ultrasound study with those of other imaging methods. It is of course clear that the purpose of the article is limited to the ultrasound study but it would be very helpful in the discussion to have a comment on the comparison with the other methods.
ANSWER: Added in paper: Comparison of US scores with other image techniques such as MRI (O-RADS MRI) could also help in the categorization of adnexal masses.
The word projection is misspelled in some places and needs to be corrected. Changed in paper
Thank you very much for your comments, they have been very useful to us

Round 2
Reviewer 3 Report
Dear Editors,
the authors have complied with the suggestions and wishes of the revision. The work has gained structure and quality as a result.
I recommend an acceptance.